# Development of human hepatocellular carcinoma in X-linked severe combined immunodeficient pigs: An orthotopic xenograft model

Kohei Mishima [1], Osamu Itano[1,2]*, Sachiko Matsuda[1], Shunichi Suzuki[3], Akira Onishi[3,4], Masashi Tamura[5], Masanori Inoue[5], Yuta Abe[1], Hiroshi Yagi[1], Taizo Hibi[6], Minoru Kitago[1], Masahiro Shinoda[1], Yuko Kitagawa[1]

1 Department of Surgery, Keio University School of Medicine, Tokyo, Japan, 2 Department of Hepato-Biliary-Pancreatic & Gastrointestinal Surgery, International University of Health and Welfare School of Medicine, Chiba, Japan, 3 Division of Animal Science, Institute of Agrobiological Sciences, National Agriculture and Food Research Organization, Ibaraki, Japan, 4 Department of Animal Science and Resources, College of Bioresource Sciences, Nihon University, Kanagawa, Japan, 5 Department of Radiology, Keio University School of Medicine, Tokyo, Japan, 6 Department of Transplantation and Pediatric Surgery, Kumamoto University Graduate School of Medical Sciences, Kumamoto, Japan

* itano@iuhw.ac.jp

**Data Availability Statement:** All relevant data are within the manuscript and its Supporting Information files.

## Abstract

Hepatocellular carcinoma (HCC) is the fifth most common primary tumor and the third leading cause of cancer-related deaths worldwide. Rodent models of HCC have contributed to the advancement of studies investigating liver carcinogenesis, tumor-host interactions, and drug screening. However, their small size renders them unsuitable for surgical or clinical imaging studies, necessitating the development of larger-size HCC models. Here, we developed a xenograft model of human HCC in X-linked interleukin-2 receptor gamma chain gene ($Il2rg$)-targeted severe combined immunodeficient (SCID) pigs. HepG2 cell suspension in serum-free medium containing 50% membrane matrix was directly injected into the liver parenchyma of eight X-linked $Il2rg$-targeted SCID pigs (6.6–15.6 kg) via ultrasonography-guided percutaneous puncture. Tumor engraftment was evaluated weekly using ultrasonography, and cone-beam computed tomography was performed during arterial portography (CTAP) and hepatic arteriography (CTHA) to evaluate the hemodynamics of engrafted tumors. The engrafted tumors were histologically analyzed following necropsy and assessed for pathological similarities to human HCCs. Macroscopic tumor formation was observed in seven of the eight pigs (simple nodular tumors in three and multinodular tumors in four). Engrafted tumors were identified as low-echoic upon ultrasonography and as perfusion-defect nodules on the CTAP images. Meanwhile, CTHA showed that the tumors were hyperattenuating. Further, histopathological findings of the engrafted tumors were consistent with those of human HCC. In conclusion, the porcine model of human HCC, successfully generated herein, might help develop more effective therapeutic strategies for HCC.

**Funding:** This work was supported by JSPS KAKENHI Grant Number JP15K06809. The funders had no role in study design, data collection and analysis, decision to publish, or preparation of the manuscript.

**Competing interests:** The authors have declared that no competing interests exist.

## Introduction

Hepatocellular carcinoma (HCC) is the fifth most common primary tumor and the third leading cause of cancer-related deaths worldwide [1]. Characterized by a high recurrence and metastasis rate, HCC is associated with poor prognosis [2, 3]. Various treatment options for HCC exist; they are selected for each patient according to international guidelines based on the number and diameter of tumors, presence or absence of distant metastases, and liver function [2, 3]. Liver resection, radiofrequency ablation (RFA), and transcatheter arterial chemoembolization (TACE) are currently available treatment options for the local control of HCC. To improve the prognoses of patients with HCC, these treatment options should be improved in addition to the development of novel therapeutic strategies. Although the use of rodent xenograft models is still central to preclinical proof-of-concept studies in experimental oncology [4, 5], considerable differences in body size compared to humans limit their utility, particularly in surgical and clinical imaging studies [6]. To overcome this limitation, studies have begun to focus on establishing large-animal xenograft models of human HCC. Accordingly, several methods have been proposed for the orthotopic transplantation of human HCC in pigs using immunosuppressive agents; however, the ensuing tumor engraftment has not yet been evaluated [7, 8]. Further, the administration of immunosuppressive agents does not sufficiently allow for the effective engraftment of human HCC cell lines into the liver parenchyma of wild-type pigs. Alternatively, several severe combined immunodeficient (SCID) pigs have been generated in the last decade, through mutagenesis or identifying natural mutations [9]. One such model is the X-linked interleukin-2 receptor gamma chain gene (*Il2rg*)-targeted SCID pigs that we first reported in 2012. *Il2rg* knockout male pigs are athymic and exhibit markedly impaired immunoglobulin and T cell and NK cell production, thus robustly recapitulating human SCID [10]. Therefore, we hypothesized that SCID pigs are an ideal platform for the implantation of human cancer cells. In this study, we aimed to develop a xenograft model of human HCC using *Il2rg*-targeted SCID pigs. To our knowledge, this is the first report of a large-animal xenograft model of human HCC.

## Materials and methods

### Animals

All animal experiments were approved by the Animal Research Ethics Committee of Keio University School of Medicine (Approval number: 20-019-9). X-linked ($Il2rg^{-/Y}$) SCID pigs were transferred from the National Agriculture and Food Research Organization to the Animal Research Center of Keio University School of Medicine. Pigs were housed individually in pathogen-free cages (1.2 m$^2$ / 22–25˚C) exposed to a 12/12-h light-dark cycle and fed a standard piglet diet along water access ad libitum. The general condition of all pigs was monitored twice a day. Orthotopic implantation was performed at the age of 1.5 months (44–49 days) (weighing 6.6–15.6 kg) (Table 1).

### Cell culture and preparation

The human HCC cell line HepG2 (ATCC, VA, USA) was used in this study. HepG2 cells were cultured in Eagle's minimum essential medium supplemented with 10% fetal bovine serum (Gibco, NY, USA) in a humidified 5% $CO_2$ incubator at 37˚C and were passaged twice per week. Before orthotopic implantation, HepG2 cells were harvested with 0.25% trypsin/EDTA (Gibco, NY, USA) and resuspended at a final concentration of $5 \times 10^7$ cells/ml in serum-free medium containing 50% membrane matrix (Corning, NY, USA).

**Table 1. Procedures performed for eight X-linked SCID pigs.**

| Pig no. | Sex | Genotype | Age at implantation (Day) | BW (kg) | HepG2 cells | Blood sampling | US | CTAP/ CTHA | Necropsy | Age at necropsy (Day) | BW (kg) |
|---|---|---|---|---|---|---|---|---|---|---|---|
| 1 | Male | IL2 RG-KO/ Y | 48 | 15.6 | $0.5 \times 10^7$ | - | + | - | + | 83 | 28.2 |
| 2 | Male | IL2 RG-KO/ Y | 48 | 14.8 | $0.5 \times 10^7$ | - | + | - | + | 83 | 30.4 |
| 3 | Male | IL2 RG-KO/ Y | 44 | 9.6 | $1 \times 10^7$ | - | + | + | + | 88 | 13.6 |
| 4 | Male | IL2 RG-KO/ Y | 44 | 8.0 | $1 \times 10^7$ | - | + | + | + | 88 | 16.0 |
| 5 | Male | IL2 RG-KO/ Y | 45 | 7.8 | $2 \times 10^7$ | + | + | + | + | 77 | 11.0 |
| 6 | Male | IL2 RG-KO/ Y | 45 | 6.6 | $2 \times 10^7$ | + | + | + | + | 77 | 9.6 |
| 7 | Male | IL2 RG-KO/ Y | 49 | 7.4 | $2 \times 10^7$ | + | + | - | + | 75 | 8.0 |
| 8 | Male | IL2 RG-KO/ Y | 49 | 9.2 | $2 \times 10^7$ | + | + | + | + | 84 | 9.0 |

BW, body weight; CTAP/CTHA, cone-beam computed tomography performed during arterial portography and hepatic arteriography; US, ultrasonography

### Anesthesia

For all procedures, pigs were anesthetized via intramuscular administration of midazolam (0.5 mg/kg) and medetomidine hydrochloride (0.01 mg/kg) combination. Thereafter, a 24-gauge intravenous catheter was set in an articular vein, and additional midazolam (0.5 mg/kg) was administered. After tracheal intubation, anesthesia was maintained using isoflurane (1.5–2.0%).

### Ectopic (subcutaneous) implantation: Preliminary experiment

Subcutaneous implantation was performed as a preliminary experiment before orthotopic implantation. A total of $1 \times 10^7$ HepG2 cells were subcutaneously injected into the abdominal wall of a 5-week-old X-linked SCID pig. The subcutaneous tissue was harvested 4 weeks later.

### Orthotopic implantation

Using a 21-gauge needle, HepG2 cell suspension ($0.5$–$2.0 \times 10^7$ cells) was injected into the liver parenchyma of each X-linked SCID pig in the supine position through an ultrasonography-guided percutaneous puncture method (Hanako Medical Co., Saitama) (Fig 1; Table 1). Under ultrasonographic guidance, the tip of the needle was placed into the liver parenchyma of the left lobe. The cells were slowly injected into the liver parenchyma, making sure there was no injury to the Glissonian pedicles or hepatic veins.

### Blood sampling

Blood samples were taken from the articular vein or internal jugular vein of X-linked SCID pigs (No. 5–8) (Table 1). Alpha-fetoprotein (AFP), aspartate aminotransferase (AST), and alanine aminotransferase (ALT) levels were measured every 7 days after the implantation of HepG2 cells.

### Ultrasonography

Whole liver ultrasonography (ALOKA, Tokyo, JPN) was performed every 7 days after implantation to assess tumor formation.

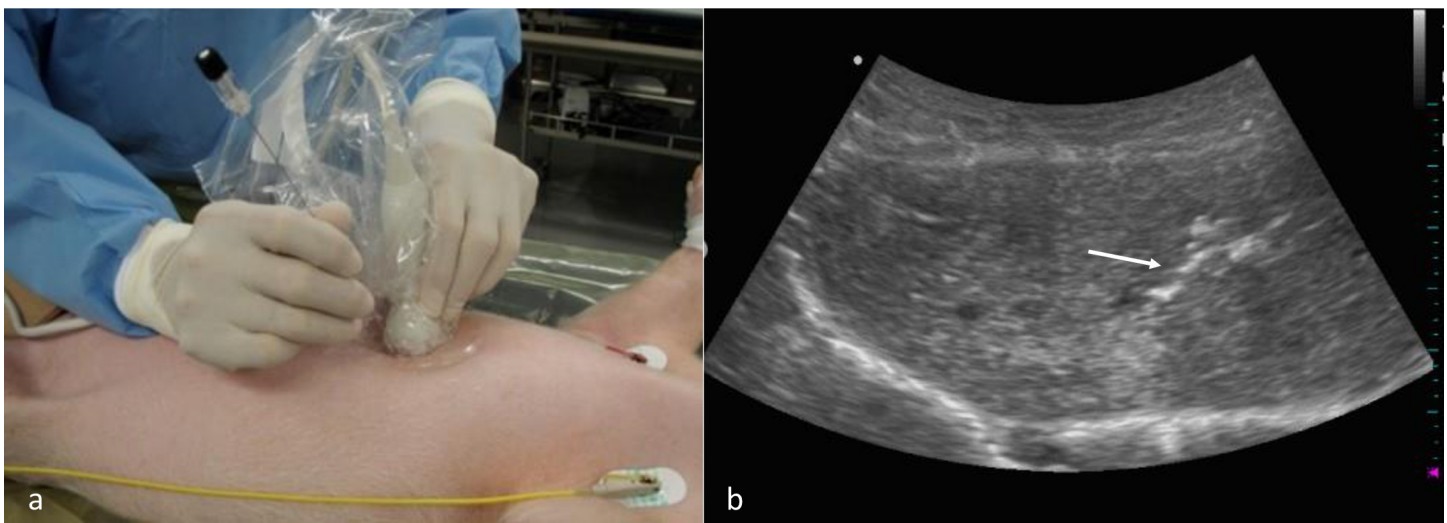

**Fig 1. Orthotopic implantation of human HCC into X-linked SCID pigs.** (a) HepG2 cells were percutaneously injected into the liver parenchyma of X-linked SCID pigs under ultrasonography guidance. (b) The tip of the needle (white arrow) was tracked throughout the procedure.

### Cone-beam computed tomography (CT) angiography

Cone-beam CT during angiography was performed on five X-linked SCID pigs (No. 3, 4, 5, 6, and 8) using a single-detector helical scanner (Table 1). With the animals in the supine position, access to the femoral artery was obtained via a 4-Fr sheath. Cone-beam CT during arterial portography (CTAP) and CT during hepatic arteriography (CTHA) were performed in combination to assess the hemodynamics of engrafted tumors. Before contrast studies, alprostadil (5 µg/body) was administered. CTAP images were obtained 40 s after initiating an intraarterial injection of 100-mL nonionic contrast medium (Oipamiron) at a rate of 5 mL/s through the superior mesenteric artery. Likewise, CTHA images were obtained 10, 30, and 50 s (1st, 2nd, and 3rd phases, respectively) after initiating an intraarterial injection of 100 ml nonionic contrast medium at the same rate through the hepatic artery.

### Necropsy

One of the pigs (No. 7) died owing to infectious gastroenteritis 26 days after orthotopic implantation. The remaining pigs were euthanized via an intravenous overdose of a commercial euthanasia solution. All the pigs were subjected to necropsy (Table 1), during which major organ systems were visually examined and macroscopic tumor formations were evaluated. Specimens were fixed with 10% formalin for 24 h and embedded in paraffin. They were then sliced into 5-µm-thick sections using a microtome and stained with hematoxylin and eosin (HE).

### Statistical analysis

All statistical analyses were performed with the SPSS statistics Version 26.0 for Windows (IBM Corp., Armonk, NY, USA). Descriptive data was compared using Chi-Square and Mann-Whitney U tests. P value less than 0.05 was considered statistically significant.

A total of eight X-linked (Il2rg)-targeted (IL2 RG-KO/Y) SCID pigs underwent orthotopic implantation of Hep G2 cells ($0.5-2 \times 10^7$ cells) between days 44 and 49. At the time of implantation, the body weight (BW) ranged from 6.6 to 15.6 kg. Blood samples were taken from four pigs (No. 5–8), ultrasonography was performed in all eight pigs every 7 days after

implantation. CTAP/CTHA was performed in five pigs (No. 3, 4, 5, 6, and 8). Necropsy was performed on all pigs between days 77 and 88 (weighing 8.0–30.4 kg).

## Results

### Ectopic (subcutaneous) implantation: Preliminary experiment

A solid tumor (1.0 × 1.0 cm) was observed in the implanted pig. The engrafted tumor was histologically consistent with moderately differentiated HCC, as revealed using HE staining (Fig 2).

### Serum markers

AFP levels increased over time in all four pigs (No. 5–8) after HepG2 cell injection. The AFP levels increased differentially in each animal, ranging from 3255 (No. 6) to 28500 (No. 8) ng/mL (2 weeks) and from 3675 (No. 6) to 548000 (No. 8) ng/mL (4 weeks). In contrast, AST and ALT levels were within normal limits throughout the experiment (Fig 3).

### Ultrasonography

Tumor engraftment was observed using ultrasonography (Fig 4; Table 2). Well-circumscribed solitary low-echoic tumors were detected in two pigs (No. 1 and 6). Mosaic pattern with halo formation was observed in two pigs (No. 3 and 5). Diffuse low-echoic lesions were observed in two pigs (No. 4 and 8).

### Angiography (CTAP and CTHA)

In pig No. 6, CTHA revealed a solitary tumor, apparent as a perfusion-defective nodule on CTAP, and ring-like enhancement on CTHA (Fig 5; Table 2). Meanwhile, multinodular tumors were observed in four pigs (No. 3, 4, 5, and 8). All tumors were observed as perfusion-defective nodules during CTAP, and CTHA revealed that they were hyperattenuating.

Tumor engraftment was confirmed with ultrasonography performed 4 weeks after HepG2 cell injection. CTAP revealed these tumors as perfusion defects and most of them appeared as hyperattenuating masses on CTHA. Macroscopic tumor formations were observed in seven of the eight pigs (simple nodular tumors in three and multinodular tumors in four).

### Gross pathology and histopathology

Tumor engraftment was observed in seven of the eight pigs. Simple nodular type tumors were observed in pig No. 1, 6, and 7, and multinodular tumors were observed in pig No. 3, 4, 5, and 8 (Fig 6). Histopathology of specimens from pigs No. 1 (Fig 7) and No. 3 (Fig 8) revealed tumor engraftment and marked tumor thrombi were observed in the portal and hepatic veins in pig No. 3 (Fig 8).

## Discussion

To the best of our knowledge, we succeeded in developing the first porcine xenograft model of human HCC in this study. Clinically, various therapeutic modalities, such as liver resection, RFA, and TACE, are utilized to treat HCC. However, HCC continues to be associated with a poor prognosis due to its tumor characteristics and underlying liver diseases [1]. Several studies have used porcine models to establish novel strategies for HCC treatment. For example, Cressman et al. [11] attempted thermoembolization with transcatheter chemistry, while Ellebrecht et al. [12] reported laparoscopic laser liver resection in normal porcine models. These

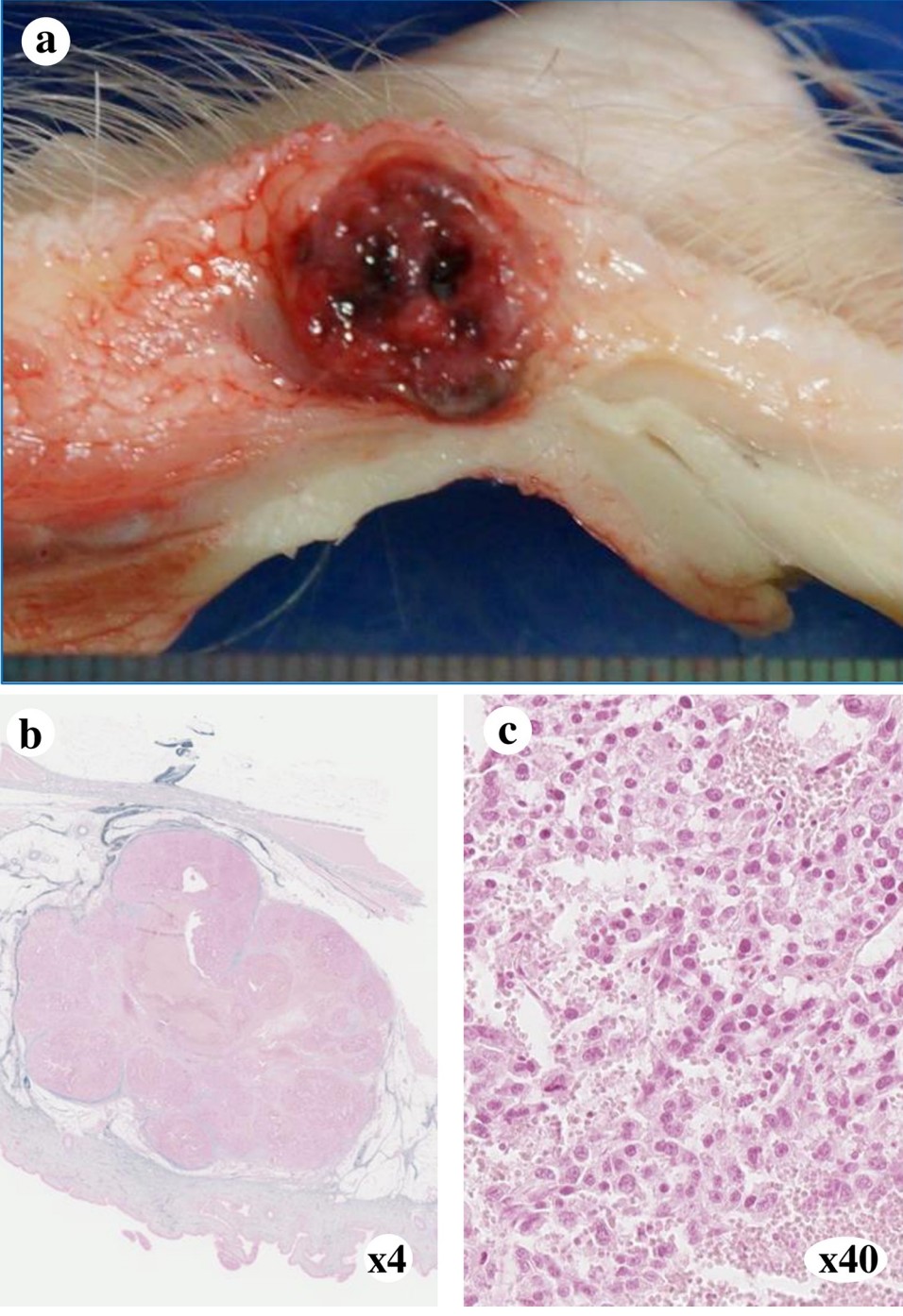

**Fig 2. Subcutaneous implantation of human HCC.** (a) Macroscopic findings of the subcutaneous tumor. An encapsulated 10 mm mass was observed. (b) Pathological findings of the subcutaneous tumor following HE staining at 4× (c) or 40× magnification. Engraftment of moderately differentiated HCC was observed.

studies concluded that due to the similar size and anatomy of pigs and humans, pigs present a useful model for testing new treatments developed via preclinical studies. Meanwhile, other studies focused on generating large-animal HCC models to evaluate the effect of local therapies

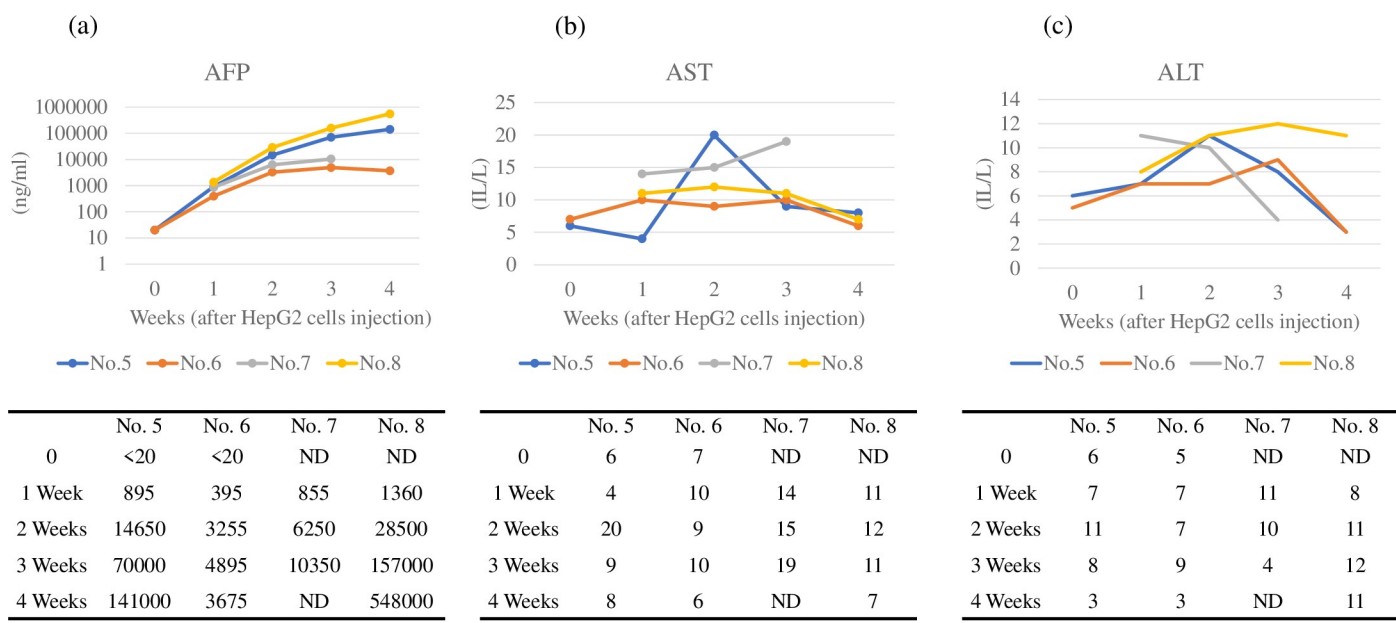

**Fig 3. Levels of serum markers in four pigs (No. 5–8).** (a) AFP levels increased over time in four pigs. (b) AST levels remained almost normal during the study. (c) ALT levels remained almost normal during the study.

or TACE. For instance, Li et al. (2006) [13], Mitchell et al. (2016) [14], and Ho et al. (2018) [15] reported diethylnitrosamine (DEN)-induced HCC in pigs, whereas Schachtschneider et al. (2017) [16] reported a translational porcine model of HCC by introducing mutations in *TP53* and *KRAS* oncogenes.

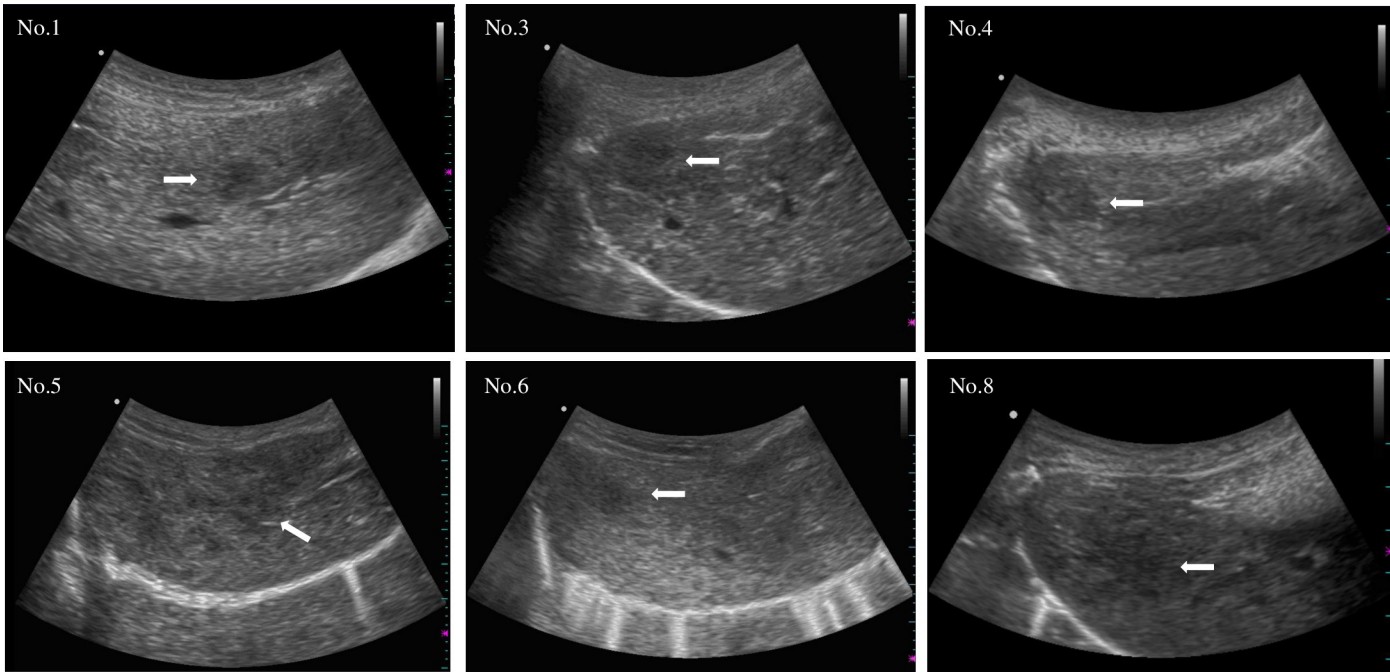

**Fig 4. Sonographic findings of the engrafted tumors 4 weeks after HepG2 cell injection.** Engrafted tumors were recognized as low-echoic lesions (white arrow).

**Table 2. Radiographic findings and gross pathology of the engrafted tumors.**

| Pig no. | Ultrasonography (4 weeks after implantation) | CTAP | CTHA | Macroscopic finding | Pathology |
|---|---|---|---|---|---|
| 1 | Low-echoic tumor | ND | ND | Simple nodular type (Solitary tumor) | HCC |
|  | Well-circumscribed |  |  |  |  |
| 2 | No tumor detected | ND | ND | No tumor engrafted | ND |
| 3 | Low-echoic tumor | Perfusion defects | Hyperattenuating | Confluent multinodular type | HCC |
|  | Unclear boundary | Multinodular tumors |  |  |  |
| 4 | Low-echoic tumor | Perfusion defects | Hypovascular | Confluent multinodular type | HCC |
|  | Unclear boundary | Multinodular tumors |  |  |  |
| 5 | Low-echoic tumor | Perfusion defects | Hyperattenuating | Confluent multinodular type | HCC |
|  | Unclear boundary | Multinodular tumors |  |  |  |
| 6 | Low-echoic tumor | Perfusion defects Solitary nodule | Ring-like enhancement | Simple nodular type (Solitary tumor) | HCC |
|  | Well-circumscribed |  |  |  |  |
| 7 | ND | ND | ND | Simple nodular type (three tumors) | HCC |
| 8 | Low-echoic tumor | Perfusion defects | Hyperattenuating | Confluent multinodular type | HCC |
|  | Unclear boundary | Multinodular tumors |  |  |  |

ND, no data; CTAP, cone-beam computed tomography performed during arterial portography; CTHA, cone-beam computed tomography performed during hepatic arteriography; HCC, hepatocellular carcinoma.

In the current study, tumor engraftment was observed in seven (simple nodular tumors in three and multinodular tumors in four) of the eight pigs. This is the first study to show the development of grossly visible human HCC in SCID pigs upon orthotopic implantation of HepG2 cells. Moreover, the engrafted tumors were well-circumscribed solid nodules, consistent with the macroscopic findings of human HCC. Cone-beam CT angiography revealed

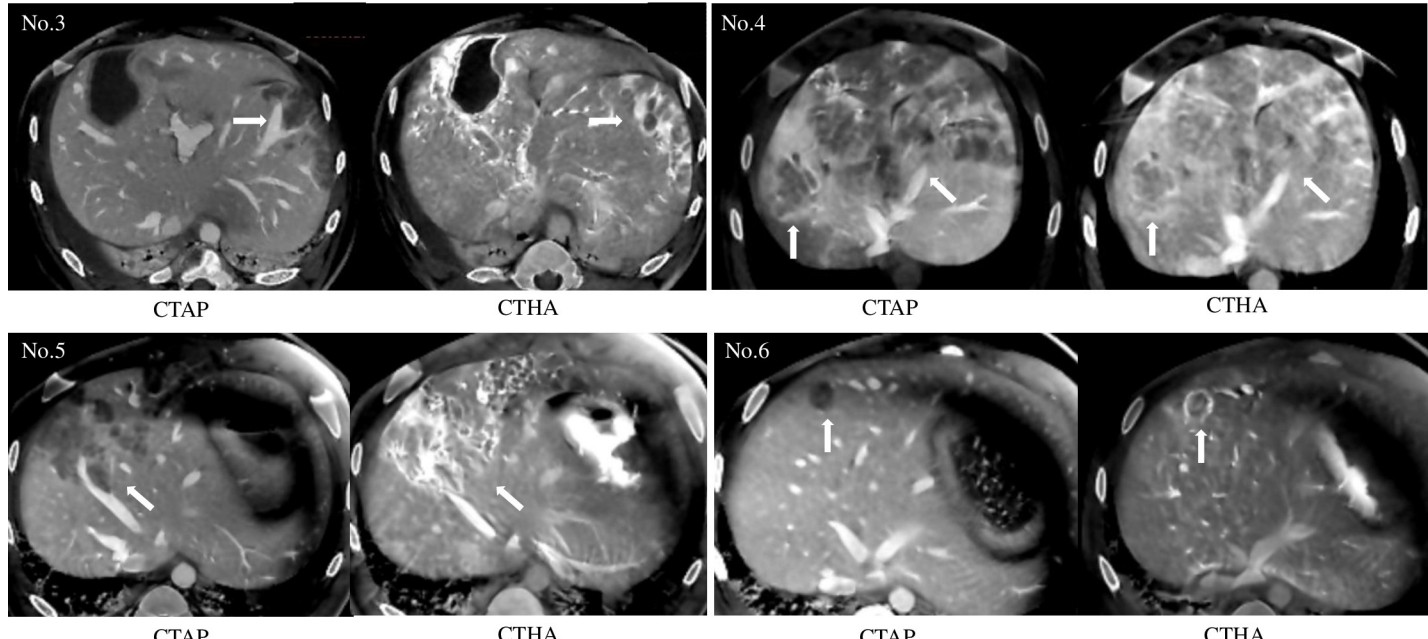

**Fig 5. Cone-beam CT angiography for engrafted tumors in pigs (No. 3–6).** (a) Tumors were observed as perfusion-defect nodules (white arrow) during CTAP. (b) CTHA revealed that tumors were hyperattenuating (white arrow).

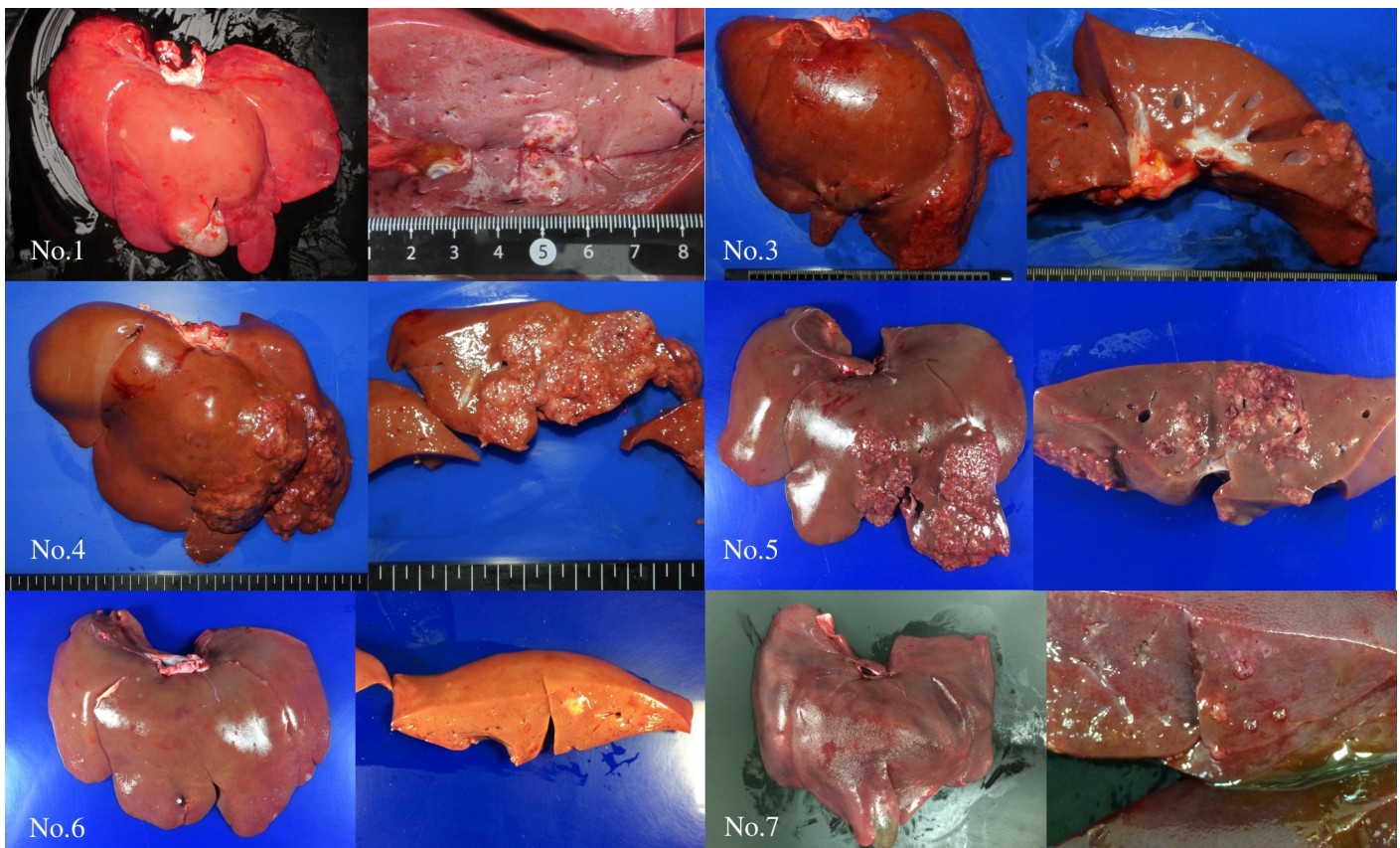

**Fig 6. Macroscopic findings of engrafted tumors in pig No. 1, 3–7.** Simple nodular type tumors were observed in pig No. 1, 6, and 7, whereas multinodular tumors were observed in pig No. 3, 4, 5, and 8.

early enhancement (hyperattenuating with CTHA) and washout (perfusion defects with CTAP), which are typical patterns of human HCC. This success in macroscopic tumor formation is presumably the result of using a porcine SCID model. Meanwhile, no macroscopic tumor formation was observed in wild pig liver parenchyma following orthotopic transplantation of human HCC cell lines with immunosuppressive agents [7].

Our model has two major advantages. First, the required duration for tumorigenesis in our model (4–6 weeks) is shorter than that in chemically induced liver tumor models (10–27 months) (13–15) (Table 3). Accordingly, our model enables the on-demand supply of HCC xenograft models and decreases the animal husbandry cost during tumorigenesis. Second, the radiographic and pathological findings of our xenograft models are consistent with those of human HCC. Since it is a human HCC (hHCC) model rather than a porcine HCC (pHCC) model, this model can be used to determine the therapeutic effect of anti-cancer drugs. The hypervascularity of the engrafted tumors is especially noteworthy, as it demonstrates that this model will be applicable for studies investigating trans-arterial chemoembolization or preclinical studies of newly developed drugs that suppress angiogenesis. Previous models included carcinogen-induced and transgenic models, most of which have longer tumor latency than the present study.

However, stabilization of the tumor engraftment in our model is an issue warranting resolution. Although procedures associated with HepG2 orthotopic implantation under

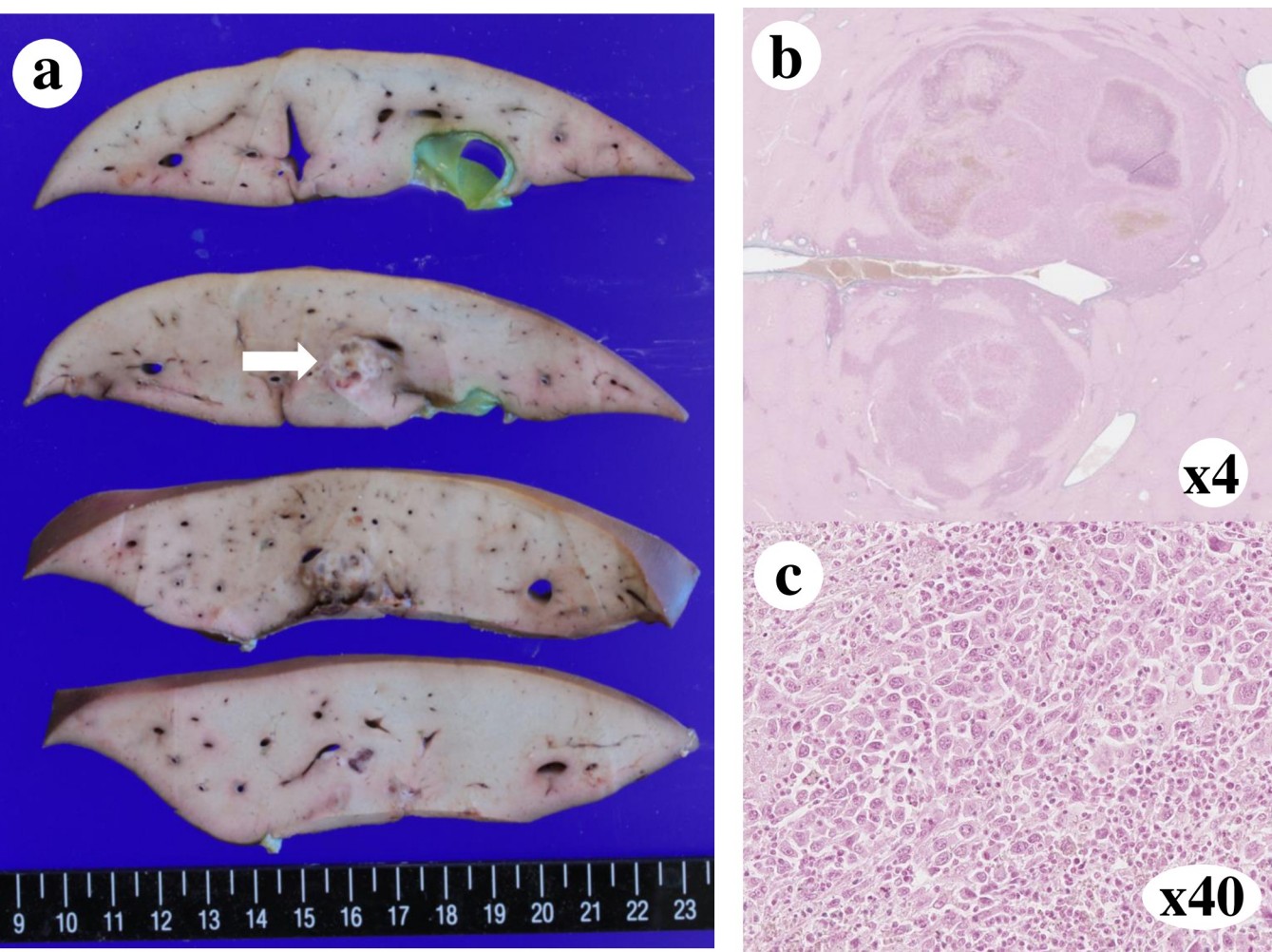

**Fig 7. Engrafted solitary tumor in the liver of pig No. 1.** (a) Macroscopic findings of the engrafted tumor. An encapsulated 10 mm mass was observed in the liver (white arrow). (b) Pathological findings of the engrafted tumor (H&E ×4). (c) Pathological findings of the engrafted tumor (H&E ×40). Engraftment of moderately differentiated HCC was observed.

ultrasonographic guidance have been standardized, the rate of tumor engraftment in the livers of the SCID pigs was 88%. This might be because the liver parenchyma did not function as a favorable microenvironment for the engraftment of implanted cells [17]. The vascularity of the liver parenchyma likely clears implanted HepG2 cells via the hepatic sinusoid and central veins into the systemic circulation [17]. Alternatively, the immune status of individual SCID pigs varies, with B cells only present until several weeks after birth in SCID pigs as they are produced in the liver during the fetal stage [10].

Controlling the number of engrafted tumors to satisfy different experimental requirements is another challenge in our model. Solitary tumors were observed in two pigs, while multiple tumors were observed in five pigs. The use of Matrigel likely improves tumor engraftment rate by preventing HepG2 cells from being discharged from the liver parenchyma [18]. However, the presence of multiple liver tumors suggested that HepG2 cells moved to other hepatic regions through the intrahepatic portal vein. Multinodular tumors exhibited a certain degree of localization and appeared randomly engrafted around the local injection site. It is unknown

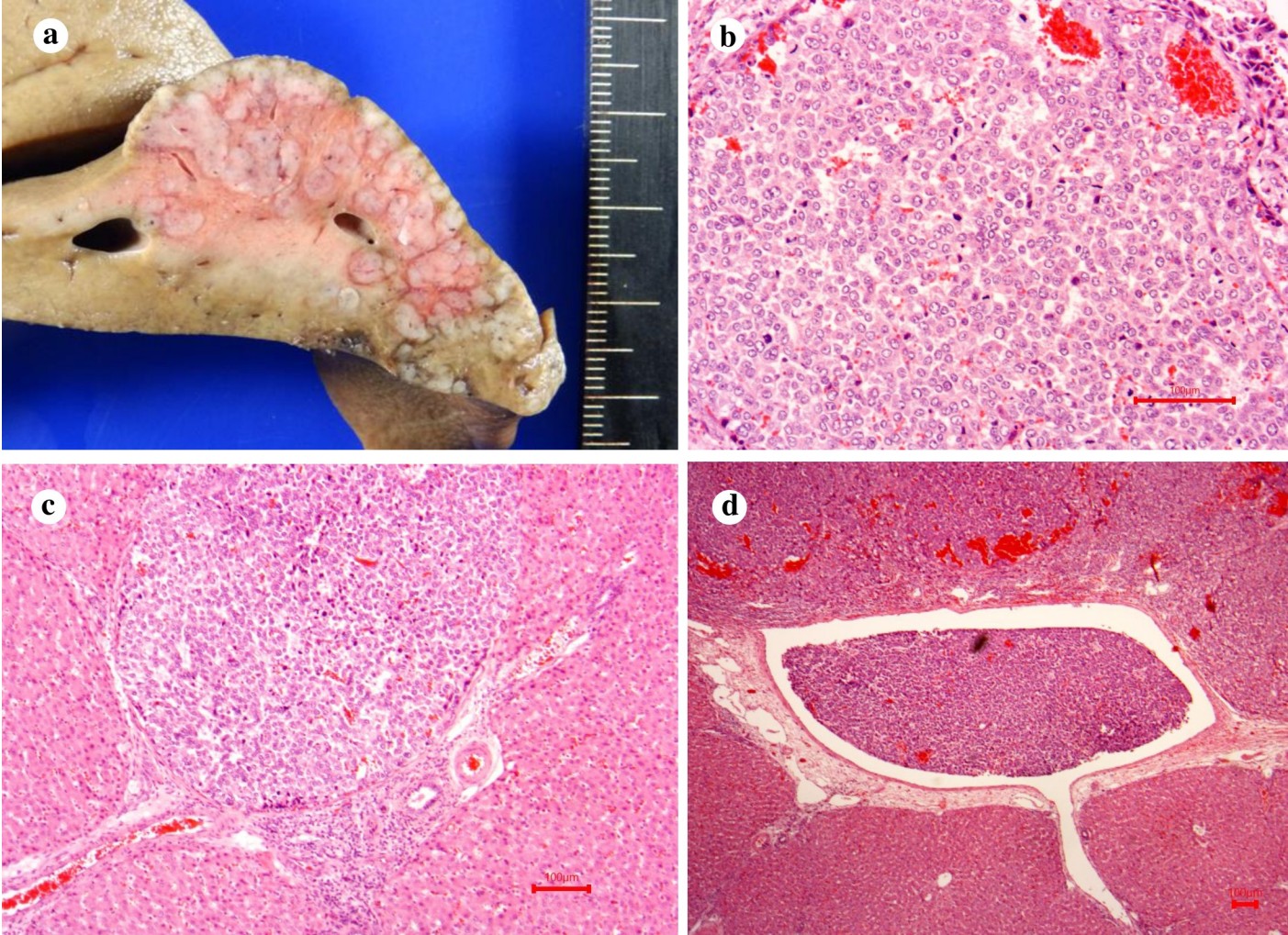

**Fig 8. Engrafted multiple tumors in the liver of pig No. 3.** (a) Macroscopic findings of the engrafted tumors. Multiple tumors were observed in the liver. (b) Pathological findings of one of the engrafted tumors (HE ×40). Engraftment of poorly differentiated HCCs was observed. Multiple tumors were observed. All tumors were observed as perfusion-defect nodules during CTAP, and CTHA showed that viable tumors were hyperattenuating. (c) Portal vein thrombus of the engrafted tumor (HE ×10). (d) Hepatic vein thrombus of the engrafted tumor (HE ×4).

whether tumor embolism in the portal vein or hepatic vein was due to the metastasis of the engrafted tumor or due to the cells entering the portal vein directly at the time of HepG2 cell injection. Implantation of a block of tumor xenograft from the subcutaneous tissue into the liver parenchyma may facilitate the development of a solitary HCC xenograft model in the future.

In summary, we developed a large-size human liver tumor xenograft model. Orthotopic implantation of human HCC cells in SCID pigs led to tumor cell engraftment and macroscopic solid tumor formation. The radiological and histopathological characteristics of the tumors were consistent with those of human HCC. The limitation of this model is the small sample size and the variation in body size among the eight animals used. Before introducing pig liver models into preclinical studies, further evaluation of cell implantation methods is required to control the tumor number and location.

**Table 3. Comparison of porcine HCC models.**

| Author | Year | Model | Treatment | N | Age at treatment | Imaging | Tumor latency | Tumor pathology |
|---|---|---|---|---|---|---|---|---|
| Li et al. | 2006 | Carcinogen-induced | DENA 10 mg/kg | 6 | 3 months | CT | 10–12 months | pHCC |
| | | | i.p. 3 months | | | MRI | | |
| Mitchell et al. | 2016 | Carcinogen-induced | DENA 15 mg/kg | 8 | 3 months | CT | 16–27 months | pHCC, HA, sarcoma |
| | | | i.p. 3 months + PLE | | | | | |
| Ho et al. | 2017 | Carcinogen-induced | DENA 15 mg/kg | 11 | 6 months | MRI | 10–18 months | pHCC, HA, FNH |
| | | | i.p. 3 months + PB 3–5 mg p.o. 4 months | | | | | |
| Schachtschneider et al. | 2017 | Transgenic | Mutations in $TP53^{R167H}$ and $KRAS^{G12D}$ | 6 | – | – | | pHCC |
| present study | | Xenograft SCID pigs | HepG2 cells | 6 | 5–6 weeks | US | 4–8 weeks | hHCC |
| | | | Percutaneous implantation | | | CT | | |

CT, computed tomography; MRI, magnetic resonance imaging; pHCC, porcine hepatocellular carcinoma; hHCC, human hepatocellular carcinoma.

## Supporting information

**S1 Table. Detailed records of the procedures performed for eight X-linked SCID pigs.**
(DOCX)

**S1 Fig. Body weight of eight X-linked SCID pigs.**
(PPTX)

**S1 Video. Procedures of ultrasonography-guided HepG2 cells implantation into the liver (pig No. 3).**
(MP4)

**S2 Video. Ultrasonographic findings of engrafted tumors in the liver at 4 weeks after implantation (pig No. 3).**
(MP4)

**S3 Video. CTAP for engrafted tumors in pig No. 3.**
(MP4)

**S4 Video. CTHA for engrafted tumors in pig No. 3.**
(MP4)

## Acknowledgments

We are grateful to Dr Fumihiko Ishikawa (RIKEN Center for Integrative Medical Sciences, Kanagawa, Japan) and the staff of Prime Tech Ltd. (Tsuchiura, Ibaraki, Japan) for their contributions to the generation of the IL2RG-knockout pig.

## Author Contributions

**Conceptualization:** Kohei Mishima, Osamu Itano, Sachiko Matsuda, Akira Onishi.

**Data curation:** Kohei Mishima, Sachiko Matsuda.

**Formal analysis:** Kohei Mishima, Osamu Itano.

**Funding acquisition:** Osamu Itano.

**Investigation:** Kohei Mishima, Osamu Itano, Sachiko Matsuda, Masashi Tamura, Masanori Inoue, Minoru Kitago.

**Methodology:** Kohei Mishima, Osamu Itano, Sachiko Matsuda, Masashi Tamura, Masanori Inoue.

**Project administration:** Osamu Itano, Sachiko Matsuda.

**Resources:** Shunichi Suzuki, Akira Onishi.

**Supervision:** Osamu Itano, Yuko Kitagawa.

**Validation:** Kohei Mishima, Osamu Itano, Yuta Abe, Hiroshi Yagi, Taizo Hibi, Minoru Kitago, Masahiro Shinoda, Yuko Kitagawa.

**Visualization:** Kohei Mishima, Sachiko Matsuda.

**Writing – original draft:** Kohei Mishima.

**Writing – review & editing:** Kohei Mishima, Osamu Itano.

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
