## [Decision Letter · Decision Letter 0]

2 Oct 2020

PONE-D-20-28805

Development of human hepatocellular carcinoma in severe combined immunodeficient pigs: An orthotopic xenograft model

PLOS ONE

Dear Dr. Osamu Itano,

Thank you for submitting your manuscript to PLOS ONE. After careful consideration, we feel that it has merit but does not fully meet PLOS ONE’s publication criteria as it currently stands. Therefore, we invite you to submit a revised version of the manuscript that addresses the points raised during the review process.  The Reviewers and the Editorial board found merit in the study.  We encourage resubmission.

Please submit your revised manuscript within 60 days. If you will need more time than this to complete your revisions, please reply to this message or contact the journal office at plosone@plos.org. Please include the following items when submitting your revised manuscript:

We look forward to receiving your revised manuscript.

Kind regards,

Gianfranco D. Alpini

Academic Editor

PLOS ONE

Journal Requirements:

2. In your Methods section, please provide additional information on the animal research and ensure you have included details on:

(i) the frequency of animal health monitoring, including the specific criteria used to monitor animal health;

(ii) Basic housing and breeding details (pathogen-free environment, light/dark cycling, cage size).

Reviewers' comments:

Reviewer's Responses to Questions

**Comments to the Author**

1. Is the manuscript technically sound, and do the data support the conclusions?

Reviewer #1: Yes

Reviewer #2: Yes

2. Has the statistical analysis been performed appropriately and rigorously? 

Reviewer #1: Yes

Reviewer #2: N/A

3. Have the authors made all data underlying the findings in their manuscript fully available?

Reviewer #1: Yes

Reviewer #2: Yes

4. Is the manuscript presented in an intelligible fashion and written in standard English?

Reviewer #1: No

Reviewer #2: Yes

5. Review Comments to the Author

Reviewer #1: In the current manuscript, Mishima et al. aimed to establish a human cell xenograft tumor model in the immunodeficient pig. The authors first tested if subcutaneous inoculation of HepG2 cells could form a tumor in the pig model. As expected, subcutaneous tumors were found in the immunodeficient pig. Thus, they did orthotopic inoculation of HepG2 cells in the pig liver. Consistently, orthotopic tumors were identified in 6 out of 8 immunodeficient pigs. Overall, the current study is simple and straightforward with signficiant contribution to the field. However, some concerns should be further clarified by the authors before its acceptance to be published in PLOS One.

1. The manuscript should be organized in a professionally scientific way.

2. The HepG2 cell is a hepatoblastoma cell line.

3. Did the authors check the pig liver function by examining serum markers? It may be helpful for scientists’ reference to compare human HCC and the porcine xenograft liver cancer model.

4. The xenograft model develops tumor foci in multiple locations. Is this due to micro-metastasis or just random engraftment?

Reviewer #2: In this study, the authors developed a novel large-animal hepatocellular carcinoma (HCC) model, which is a xenograft model of human HCC in the X-linked interleukin-2 receptor gamma chain gene (Il2rg)-targeted severe combined immunodeficient (SCID) pigs and aimed to use this model for surgical or clinical imaging studies on HCC. HepG2 cell suspension was directly injected into the liver parenchyma of Il2rg-targeted SCID pigs via ultrasonography-guided percutaneous puncture. The authors demonstrated that the successfully generated porcine model of human HCC might enable developing better therapeutic strategies for HCC. Overall, data provided here by the authors could have interesting implications. Specific points need to be considered are listed below:

1. To better characterize the orthotopic xenograft model, the authors need to measure the serum levels of ALT, AST, etc.

2. It’s necessary to mention the differences between pHCC and hHCC.

3. Liver weight and liver to body weight ratio should be included in Table 1.

4. The species of the pigs, as well as liver pathology need to be included in Table 3.

5. The authors need to mention the gender of the pigs used for the xenograft.

6. PLOS authors have the option to publish the peer review history of their article (what does this mean?). If published, this will include your full peer review and any attached files.

Reviewer #1: No

Reviewer #2: No

---

## [Author Response · Author response to Decision Letter 0]

13 Feb 2021

February 14, 2021

Gianfranco D. Alpini

Academic Editor

PLOS ONE

Dear Editor:

I wish to re-submit the revised version of our article for publication in PLOS ONE, titled “Development of human hepatocellular carcinoma in X-linked severe combined immunodeficient pigs: An orthotopic xenograft model.” The paper was co-authored by Kohei Mishima, Sachiko Matsuda, Shunichi Suzuki, Akira Onishi, Masashi Tamura, Masanori Inoue, Yuta Abe, Hiroshi Yagi, Taizo Hibi, Minoru Kitago, Masahiro Shinoda, and Yuko Kitagawa. The manuscript ID is PONE-D-20-28805.

We are thankful for the constructive comments, which have helped us to considerably improve the overall presentation of our manuscript. We have provided a point-by-point response to each reviewer comment below. 

We hope that the changes incorporated into the revised manuscript satisfactorily address the reviewers’ concerns and our manuscript will now be considered suitable for publication in your journal.

We thank you for your consideration and look forward to hearing from you. 

Sincerely,

Professor Osamu Itano

Department of Hepato-Biliary-Pancreas & Gastrointestinal Surgery, 

International University of Health and Welfare School of Medicine, 

Chiba, 286-8686, Japan. 

Email: itano@iuhw.ac.jp

Journal Requirements:

Response: We have re-edited the entire manuscript to ensure that it meets your style requirements. English proofreading was performed by Editage (www.editage.jp), a company that provides professional English language editing services..

2. In your Methods section, please provide additional information on the animal research and ensure you have included details on:

(i) the frequency of animal health monitoring, including the specific criteria used to monitor animal health;

(ii) Basic housing and breeding details (pathogen-free environment, light/dark cycling, cage size).

Response: We have added several sentences to the Material and Methods section to address the points raised (Page 6, Line 83-87). 

Response: We have deleted the ethics statement from other sections of the manuscript apart from the methods.

Comments from reviewers:

Reviewer #1: In the current manuscript, Mishima et al. aimed to establish a human cell xenograft tumor model in the immunodeficient pig. The authors first tested if subcutaneous inoculation of HepG2 cells could form a tumor in the pig model. As expected, subcutaneous tumors were found in the immunodeficient pig. Thus, they did orthotopic inoculation of HepG2 cells in the pig liver. Consistently, orthotopic tumors were identified in 6 out of 8 immunodeficient pigs. Overall, the current study is simple and straightforward with significant contribution to the field. However, some concerns should be further clarified by the authors before its acceptance to be published in PLOS One.

1. The manuscript should be organized in a professionally scientific way.

Response: Along with English proofreading, the entire manuscript has been polished for a more scientific tone. Further, we have added radiographic data of US, CT, and macroscopic findings of SCID pigs (Fig 4, 5, and 6).

2. The HepG2 cell is a hepatoblastoma cell line.

Response: In addition to the hepatoblastoma cell line (PubMed: 19751877), HepG2 is also referred as HCC cell line based on the original publication (PubMed: 6248960). According to the ATCC (VA, USA, https://www.lgcstandards-atcc.org/en/Products/All/HB-8065.aspx), HepG2 was derived from the HCC of a 15-year-old Caucasian male. 

3. Did the authors check the pig liver function by examining serum markers? It may be helpful for scientists’ reference to compare human HCC and the porcine xenograft liver cancer model.

Response: We performed blood sampling in four pigs (No. 5-8). AFP levels increased over time after HepG2 cell injection and this has been shown in Figure 3.

4. The xenograft model develops tumor foci in multiple locations. Is this due to micro-metastasis or just random engraftment?

Response: As shown in the Figure 5 (CT images) and 6 (Macroscopic findings), multinodular tumors had a certain degree of localization and appeared randomly engrafted around the local injection site. It is unknown whether tumor embolism in the portal vein or hepatic vein was due to the metastasis of the engrafted tumor or due to the HepG2 cells entering the portal vein directly at the time of HepG2 cell injection (Page 19, Line 296-300). 

Reviewer #2: In this study, the authors developed a novel large-animal hepatocellular carcinoma (HCC) model, which is a xenograft model of human HCC in the X-linked interleukin-2 receptor gamma chain gene (Il2rg)-targeted severe combined immunodeficient (SCID) pigs and aimed to use this model for surgical or clinical imaging studies on HCC. HepG2 cell suspension was directly injected into the liver parenchyma of Il2rg-targeted SCID pigs via ultrasonography-guided percutaneous puncture. The authors demonstrated that the successfully generated porcine model of human HCC might enable developing better therapeutic strategies for HCC. Overall, data provided here by the authors could have interesting implications. Specific points need to be considered are listed below:

1. To better characterize the orthotopic xenograft model, the authors need to measure the serum levels of ALT, AST, etc.

Response: Frozen sera from four pigs were used and the timeline of AST, ALT, and AFP level monitoring is shown in Figure 3. Although no changes were observed in AST and ALT levels, an increase in AFP levels reflected tumor engraftment. 

2. It’s necessary to mention the differences between pHCC and hHCC.

Response: In the Discussion section, we have included a sentence mentioning differences between human HCC (hHCC) and porcine HCC (pHCC) (Page 17, Line 270-271).

3. Liver weight and liver to body weight ratio should be included in Table 1.

Response: Unfortunately, we did not measure liver weight in this study.

4. The species of the pigs, as well as liver pathology need to be included in Table 3.

Response: We have added genotypic information in Table 1. Information on liver pathology has been added to Table 2.

5. The authors need to mention the gender of the pigs used for the xenograft.

Response: We have included gender information in Table 1. We have also included information on age, body weight, and genotype.

---

## [Decision Letter · Decision Letter 1]

25 Feb 2021

Development of human hepatocellular carcinoma in X-linked severe combined immunodeficient pigs: An orthotopic xenograft model

PONE-D-20-28805R1

Dear Dr. Osamu Itano,

We’re pleased to inform you that your manuscript has been judged scientifically suitable for publication and will be formally accepted for publication once it meets all outstanding technical requirements.

Kind regards,

Gianfranco D. Alpini

Academic Editor

PLOS ONE

Additional Editor Comments (optional):

Reviewers' comments:

Reviewer's Responses to Questions

**Comments to the Author**

1. If the authors have adequately addressed your comments raised in a previous round of review and you feel that this manuscript is now acceptable for publication, you may indicate that here to bypass the “Comments to the Author” section, enter your conflict of interest statement in the “Confidential to Editor” section, and submit your "Accept" recommendation.

Reviewer #1: All comments have been addressed

Reviewer #2: All comments have been addressed

2. Is the manuscript technically sound, and do the data support the conclusions?

Reviewer #1: Yes

Reviewer #2: Yes

3. Has the statistical analysis been performed appropriately and rigorously? 

Reviewer #1: Yes

Reviewer #2: Yes

4. Have the authors made all data underlying the findings in their manuscript fully available?

Reviewer #1: Yes

Reviewer #2: Yes

5. Is the manuscript presented in an intelligible fashion and written in standard English?

Reviewer #1: Yes

Reviewer #2: Yes

6. Review Comments to the Author

Reviewer #1: Thanks for your responses. The authors have carefully addressed all my concerns. The revised manuscript is acceptable.

Reviewer #2: (No Response)

7. PLOS authors have the option to publish the peer review history of their article (what does this mean?). If published, this will include your full peer review and any attached files.

Reviewer #1: No

Reviewer #2: No

---

## [Editor Report · Acceptance letter]

11 Mar 2021

PONE-D-20-28805R1 

Development of human hepatocellular carcinoma in X-linked severe combined immunodeficient pigs: An orthotopic xenograft model 

Dear Dr. Itano:

I'm pleased to inform you that your manuscript has been deemed suitable for publication in PLOS ONE. Congratulations! Your manuscript is now with our production department. 

Kind regards, 

on behalf of

Dr. Gianfranco D. Alpini 

Academic Editor

PLOS ONE